# Mitigating Mode Collapse in Sequential Disentanglement via An Architecture Bias

## Abstract

One of the fundamental representation learning tasks is unsupervised sequential disentanglement, where latent representations of the inputs are decomposed to a single static factor and a sequence of dynamic factors. To extract this latent information, existing variational methods condition the static and dynamic codes on the entire input sequence. Unfortunately, these models often suffer from *mode collapse*, i.e., the dynamic vectors encode static and dynamic information, leading to a non-meaningful static component. Attempts to alleviate this problem via reducing the dynamic dimension and mutual information loss terms gain only partial success. Often, promoting a certain functionality of the model is better achieved via specific architectural biases instead of incorporating additional loss terms. For instance, convolutional nets gain translation-invariance with shared kernels and attention models realize the underlying correspondence between source and target sentences. Inspired by these successes, we propose in this work a novel model that mitigates mode collapse by conditioning the static component on a *single* sample from the sequence, and subtracting the resulting code from the dynamic factors. Remarkably, our variational model has less hyper-parameters in comparison to existing work, and it facilitates the analysis and visualization of disentangled latent data. We evaluate our work on multiple data-modality benchmarks including general time series, video, and audio, and we show beyond state-of-the-art results on generation and prediction tasks in comparison to several strong baselines.

## 1 Introduction

Modern representation learning (Bengio et al., 2013; Schölkopf et al., 2021) identifies unsupervised disentanglement as one of its fundamental challenges, where the main goal is to decompose input data to its latent factors of variation. Disentangled representations are instrumental in numerous applications including classification (Locatello et al., 2020), prediction (Hsieh et al., 2018), and interpretability (Higgins et al., 2016), to name just a few. In the sequential setting, input sequences are typically split to a single static (time-invariant) factor encoding features that do not change across the sequence, and to multiple dynamic (time-varying) components, one per sample.

Existing sequential disentanglement works are commonly based on variational autoencoders (VAEs) Kingma & Welling (2014) and their dynamic extensions (Girin et al., 2021). To model the variational posterior, Li & Mandt (2018); Bai et al. (2021) and others condition the static and dynamic factors on the entire input sequence. However, under this modeling perspective, highly expressive deep encoder modules struggle with *mode collapse* problems. Namely, the learned time-varying components capture dynamic as well as static information, whereas the static vector encodes non-meaningful features. To resolve this issue, Li & Mandt (2018) propose to reduce the dynamic dimension, and Zhu et al. (2020); Bai et al. (2021) introduce additional mutual information (MI) loss terms. While these approaches generally improved disentanglement and mode collapse issues, new challenges emerged. First, small time-varying vectors are limited in modeling complex dynamics. Second, sampling good positive and negative examples to estimate mutual information is difficult (Naiman et al., 2023). Finally, optimizing models with multiple losses and balancing their three MI penalties is computationally expensive. These challenges raise the question: can we model and implement unsupervised sequential disentanglement that avoids mode collapse without additional loss terms or reducing dimensionality?

To induce deep neural networks with a certain behavior, prior successful attempts opted for designing tailored architectures, often arising from an overarching assumption. For instance, convolutional neural networks (LeCun et al., 1989) use shared kernel filters across natural images, exploiting their translation-invariance structure. Similarly, natural language processing methods (Bahdanau et al., 2015) employ attention modules, assuming not all words in the source sentence share the same effect on the target sentence. Inspired by these successes, we suggest a novel sequential disentanglement model that drastically alleviates mode collapse, based on the assumption that the static posterior can be conditioned on a *single* sample from the sequence. Further, our modeling guides architecture design, where we learn the static factor separately from the sequence, and we subtract its contents from the learned dynamic factors series. The resulting framework has no MI terms and thus less hyper-parameters, and further, it requires no constraints on the dimension of factors.

The contributions of our work can be summarized as follows:

1. We introduce a novel sequential disentanglement model whose static posterior is conditioned on a single series element, leading to a new neural architecture that learns the static factor independently of the sequence, and subtracts its contents from the dynamic factors.

2. Our method do not restrict the disentangled components' dimension, allowing to learn complex dynamics. Further, it mitigates mode collapse without mutual information loss terms, yielding a simple training objective with only two hyper-parameters.

3. We extensively evaluate our approach both qualitatively and quantitatively on several data modalities including general time series, video and audio. We obtain state-of-the-art results on standard challenging benchmarks in comparison to several strong baseline techniques.

## 2    RELATED WORK

**Unsupervised disentanglement.** A large body of work is dedicated to studying unsupervised disentanglement using VAEs (Kingma & Welling, 2014). For instance, Higgins et al. (2016) augment VAEs with a hyper-parameter weight on the Kullback–Liebler divergence term, leading to improved disentanglement of latent factors. In Kumar et al. (2018), the authors regularize the expectation of the posterior, and Bouchacourt et al. (2018) learn multi-level VAE by grouping observations. Kim & Mnih (2018) promote independence across latent dimensions, and Chen et al. (2018) decompose the objective function and identify a total correlation term. In addition, generative adversarial networks (GANs) are used to maximize mutual information (Chen et al., 2016) and to compute an attribute dependency metric (Wu et al., 2021).

**Sequential disentanglement.** Early probabilistic models suggest conditioning on the mean of past features (Hsu et al., 2017), and directly on the features Li & Mandt (2018). Disentangling video sequences is achieved using generative adversarial networks (Villegas et al., 2017; Tulyakov et al., 2018) and a recurrent model with an adversarial loss (Denton & Birodkar, 2017). To address mode collapse issues, Zhu et al. (2020) introduce modality-specific auxiliary tasks and supervisory signals, whereas Bai et al. (2021) utilize contrastive estimation using positive and negative examples. Optimal transport and the Wasserstein distance are used to regularize the evidence lower bound in (Han et al., 2021). Tonekaboni et al. (2022) extract local and global representations to encode non-stationary time series data. Recently, Naiman et al. (2023) developed a modality-independent approach that samples positive and negative examples directly from latent space. Others considered the more general multifactor sequential disentanglement, where every sequence is decomposed to multiple static and dynamic factors (Bhagat et al., 2020; Yamada et al., 2020; Berman et al., 2023).

## 3    BACKGROUND

**Problem formulation.** We generally follow the notation and terminology introduced in (Li & Mandt, 2018). Let $x_{1:T} = \{x_1, \ldots, x_T\}$ denote a multivariate sequence of length $T$, where $x_t \in \mathbb{R}^d$ for every $t$. Given a dataset $\mathcal{D} = \{x_{1:T}^j\}_{j=1}^N$, the goal of sequential disentanglement is to extract an alternative representation of $x_{1:T}$, where we omit $j$ for brevity, via a static (time-invariant) factor $s$ and multiple dynamic (time-varying) components $d_{1:T}$.

**Sequential probabilistic modeling.** Typically, the static and dynamic features are assumed to be independent, and thus the joint distribution is given by

$$p(x_{1:T}, z \,;\, \theta, \psi) = \left[ p(s) \prod_{t=1}^{T} p(d_t \,|\, d_{<t} \,;\, \psi) \right] \cdot \prod_{t=1}^{T} p(x_t \,|\, s, d_t \,;\, \theta) \,, \tag{1}$$

where $z := (s, d_{1:T})$ combines static and dynamic components, $d_t$ depends on prior features $d_{<t}$, and $x_t$ can be reconstructed from the static and current dynamic codes. The static prior distribution is modeled by a standard Gaussian distribution $p(s) := \mathcal{N}(0, I)$, whereas the dynamic prior is computed via a recurrent neural network $p(d_t \,|\, d_{<t} \,;\, \psi) := \mathcal{N}(\mu(d_{<t}), \sigma^2(d_{<t}) \,;\, \psi)$ to capture nonlinear dynamics (Chung et al., 2015). Exploiting the independence between time-varying and time-invariant factors, the approximate posterior is parameterized by $\phi = (\phi_s, \phi_d)$, and it reads

$$q(s, d_{1:T} \,|\, x_{1:T} \,;\, \phi) = q(s \,|\, x_{1:T} \,;\, \phi_s) \prod_{t=1}^{T} q(d_t \,|\, d_{<t}, x_{\leq t} \,;\, \phi_d) \,. \tag{2}$$

**Objective function.** The corresponding evidence lower bound related to Eqs. 1 and 2 is given by

$$\max_{\theta, \phi, \psi} \mathbb{E}_{p_{\mathcal{D}}} \left[ \mathbb{E}_{z \sim q_\phi} \log p(x_{1:T} \,|\, z \,;\, \theta) - \beta \, \mathrm{KL}[q(z \,|\, x_{1:T} \,;\, \phi) \,\|\, p(z \,;\, \psi)] \right] \,, \tag{3}$$

where $p_{\mathcal{D}}$ is the train set distribution, $q_\phi := q(z \,|\, x_{1:T} \,;\, \phi)$, and $\beta \in \mathbb{R}^+$.

Unfortunately, the described model above is prone to mode collapse. Namely, the posterior code $d_t$ may capture the static *and* dynamic factors of variation in $x_t$, for every $t$, resulting in non-meaningful static codes $s$. To alleviate this problem, Li & Mandt (2018) suggest to decrease the dimension of $d_t$ such that $\dim(d_t) \ll \dim(s)$. The intuition behind this heuristic is to provide the model with a minimal subspace for learning dynamics, without extra degrees of freedom to capture the static information. Other approaches (Zhu et al., 2020; Bai et al., 2021) aim to mitigate mode collapse by augmenting the objective function with mutual information terms, minimizing $I_q(s; d_{1:T})$ and maximizing $I_q(s; x_{1:T})$ and $I_q(d_{1:T}; x_{1:T})$. However, evaluating $I_q(u; v)$ is difficult (Chen et al., 2018), and it is approximated by contrastive estimation (Oord et al., 2018) which requires positive and negative data samples. Overall, while the above solutions lessen the detrimental effects of mode collapse, several major challenges still exist. 1) the hyper-parameter $\dim(d_t)$ may be difficult to tune, and it limits model expressivity in capturing complex dynamics; 2) contrastive estimation is often implemented with domain-dependent data augmentation, hindering its use on arbitrary data modalities (Tonekaboni et al., 2022); and 3) models with multiple loss terms (including MI) may be sensitive to hyper-parameter choices and their optimization is computationally expensive.

## 4 Method

Motivated by the challenges described in Sec. 3, our main goal is to answer the question:

*"Can we learn modality-independent disentangled representations while alleviating mode collapse by a variational model whose implementation can be realized via an architecture design?"*

To this end, we make the following two observations: (i) the approximate posterior distribution is ultimately responsible for extracting latent factors of variation from input sequences, and thus, mode collapse issues mainly appear in the posterior; and (ii) the static features are, by definition, time-invariant and shared across the sequence, and therefore, they could be extracted from a single sample in the sequence. Based on these simple observations, we propose a new posterior distribution that conditions the static factor on one sequential element, e.g., the first item $x_1$. Intuitively, $x_1$ may be viewed as an *anchor* example, from which we extract the static information, and we set its corresponding dynamic factor to be the zero vector, i.e., $d_1 := 0$. Moreover, we assume that the time-invariant factor is required in modeling the time-varying components. Formally, our posterior distribution is defined as follows

$$q(s, d_{1:T} \,|\, x_{1:T} \,;\, \phi) = q_{\phi_s} \cdot q_{\phi_d} := q(s \,|\, x_1 \,;\, \phi_s) \cdot \prod_{t=2}^{T} q(d_t \,|\, s, d_{<t}, x_{\leq t} \,;\, \phi_d) \,, \tag{4}$$

where we set $d_1 := 0$ and similarly to Eq. 2, $\phi = (\phi_s, \phi_d)$, that is, the static and dynamic codes are computed using neural networks parameterized by $\phi_s$ and $\phi_d$, respectively.

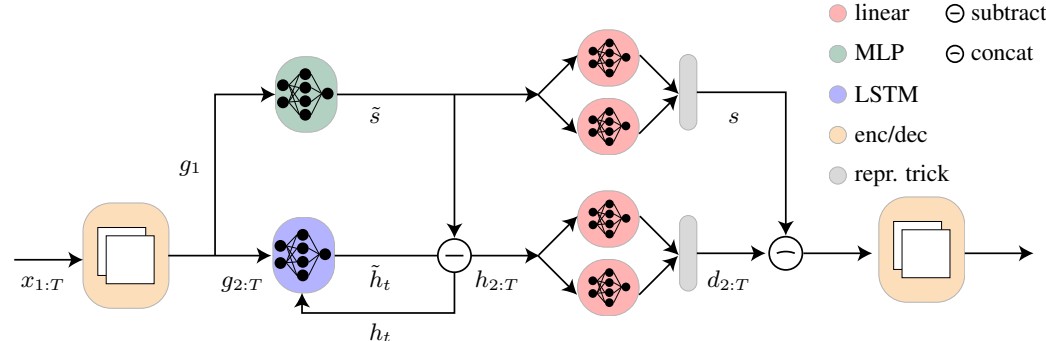

Figure 1: Our network is composed of an encoder (left), a decoder (right) and two paths in-between for computing the static factor (top) and the dynamic components (bottom).

In addition to the posterior above, our model uses the same prior distribution as in Eq. 1. Further, given the unique role of $x_1$ in our posterior, we also need to describe the modifications to the loss function in Eq. 3. Specifically, the reconstruction term $\mathbb{E}_{z \sim q_\phi} \log p(x_{1:T} \mid z\,;\,\theta)$ is split to

$$\mathcal{L}_{\text{recon}} = \mathbb{E}_{s \sim q_{\phi_s}} \left[ \mathbb{E}_{d_{2:T} \sim q_{\phi_d}} \log p(x_{2:T} \mid s, d_{2:T}\,;\,\theta) + \alpha \, \log p(x_1 \mid s, 0\,;\,\theta) \right]\,, \qquad (5)$$

where, notably, $\alpha \in \mathbb{R}^+$ could potentially be taken as 1. However, we typically obtain better results in practice when $\alpha \neq 1$. The regularization KL term can be elaborated as follows

$$\mathcal{L}_{\text{reg}} = \beta \, \text{KL}[q(s \mid x_1\,;\,\phi_s) \,\|\, p(s)] + \beta \, \text{KL}[q(d_{2:T} \mid x_{2:T}\,;\,\phi_d) \,\|\, p(d_{2:T}\,;\,\psi)]\,, \qquad (6)$$

with $\beta \in \mathbb{R}^+$, $q(d_{2:T} \mid x_{2:T}\,;\,\phi_d) = q_{\phi_d}$ and $p(d_{2:T}\,;\,\psi) = \prod_{t=2}^{T} p(d_t \mid d_{<t}\,;\,\psi)$ via Eq. 1. Overall, when combining Eqs. 5 and 6, we arrive at the following total objective function

$$\mathcal{L} = \max_{\theta, \phi, \psi} \mathbb{E}_{p_D}(\mathcal{L}_{\text{recon}} + \mathcal{L}_{\text{reg}})\,. \qquad (7)$$

**An architectural bias.** Our sequential disentanglement deep neural network takes the sequence $x_{1:T}$ as input, and it returns its reconstruction as output. The inputs are processed by a problem dependent encoder, yielding an intermediate sequential representation $g_{1:T}$. Notice, that this module is not recurrent, and thus every $x_t$ is processed independently of other $x_u$ where $u \neq t$. The sequence $g_{1:T}$ is split into two parts $g_1$ and $g_{2:T}$, undergoing two paths. To extract the static information, $g_1$ is passed to a simple multilayer perceptron (MLP) consisting of a linear layer and `tanh` as activation, producing $\tilde{s}$. Then, two linear layers learn the mean $\mu(\tilde{s})$ and variance $\sigma^2(\tilde{s})$, allowing to sample the static factor $s$ via the reparametrization trick. To extract the dynamic information, $g_{2:T}$ is fed to a long short-term memory (LSTM) module (Hochreiter & Schmidhuber, 1997) with a hidden state $\tilde{h}_t$ we use to define a new hidden state $h_t$ by subtracting $\tilde{s}$. Formally,

$$h_t := \tilde{h}_t - \tilde{s}\,, \quad t = 2, \ldots, T\,. \qquad (8)$$

Subtracting $\tilde{s}$ mitigates mode collapse by "removing" static features that exist in $\tilde{h}_t$, and thus limiting the ability of the LSTM module to extract time-invariant information. We use $h_{2:T}$ to compute $\mu(h_t)$ and $\sigma^2(h_t)$ using two linear layers, and we sample $d_{2:T}$ by the reparametrization trick. The static and dynamic factors are combined $(s, d_t)$ and passed to a domain-dependent decoder, to produce the reconstruction of $x_{1:T}$. Please see Fig. 1 for a graphical illustration and notation of our model.

## 5 EXPERIMENTS

### 5.1 EXPERIMENTAL SETUP

We perform an extensive qualitative, quantitative, and ablation evaluation of our method on datasets of different modalities: video sequences, general time series, and audio recordings. For video sequences, we use the **Sprites** dataset that contains moving cartoon characters (Reed et al., 2015), and the **MUG** dataset that includes several subjects with various facial expressions (Aifanti et al., 2010).

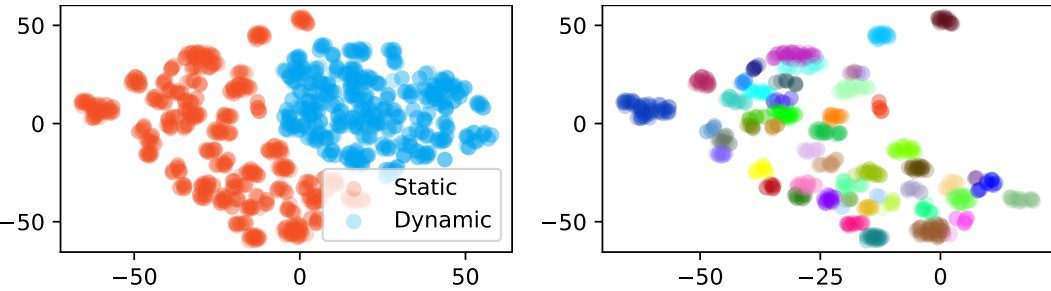

Figure 2: t-SNE plots on MUG dataset of the latent static and dynamic factors (left), and only latent static codes, colored by subject identity (right).

The general time series datasets are **PhysioNet**, consisting of medical records of patients (Goldberger et al., 2000), **Air Quality** which includes measurements of air pollution (Zhang et al., 2017), and **ETTh1** that measures the electricity transformer temperature (Zhou et al., 2021). For audio recordings, we consider the **Timit** datasets, consisting of read speech of short sentences (Garofolo, 1993). The evaluation tests and tasks we consider serve as the standard benchmark for sequential disentanglement. We compare our approach to recent state-of-the-art (SOTA) techniques including FHVAE (Hsu et al., 2017), DSVAE (Li & Mandt, 2018), MoCoGan (Tulyakov et al., 2018), R-WAE (Han et al., 2021), S3VAE (Zhu et al., 2020), C-DSVAE (Bai et al., 2021), SKD (Berman et al., 2023), GLR (Tonekaboni et al., 2022), and SPYL (Naiman et al., 2023). See the appendices for more details regarding datasets (App. A.1), evaluation metrics (App. A.2), training hyper-parameters (App. A.3), and neural architectures (App. A.4).

## 5.2 QUALITATIVE EVALUATION

In what follows, we qualitatively assess our approach on video sequences and general time series data. To this end, we analyze the learned latent space of the static and dynamic factors of variation by utilizing the dimensionality reduction technique, t-distributed Stochastic Neighbor Embedding (t-SNE) (Van der Maaten & Hinton, 2008). In addition, we also swap factors of variation between two separate sequences, similar to e.g., (Li & Mandt, 2018).

**Static and dynamic clustering.** To assess the capacity of our approach to disentangle data into distinct subspaces, we perform the following: First, we randomly select a batch of samples from a given test set. Second, we extract the static $s^j$ and dynamic $d^j_{1:T}$ latent disentangled representations for each sample $j$ from the batch. Third, we concatenate the dynamic factors into a single vector, i.e., $d^j := \text{concat}(d^j_{1:T})$. Finally, we use t-SNE to project the collection of static and dynamic representations into a two-dimensional embedding space. We visualize the obtained embedding of MUG dataset in Fig. 2)(left), where static factors are colored in blue and the dynamic factors in orange. The results show that our approach clearly clusters between the time-varying and time-invariant factors of variation. Further, we can observe distinct sub-clusters within the orange point cloud, which may indicate a hierarchical clustering based on the identities of subjects. To verify this assumption, we plot the static embedding colored by subject in Fig. 2(right). We omitted the legend as it clutters the result. Indeed, our model learns a clustered representation with respect to people's identities without any explicit constraints. Finally, we repeated this experiment and we present the static and dynamic t-SNE embeddings of three time series datasets in Fig. 6. Similarly to MUG, we observe a clear separation between disentangled codes in all cases.

**Sequential swap.** We will now show the ability of our model to *swap* between static and dynamic representations of two different sequences. Specifically, we take two sequences $x^1_{1:T}, x^2_{1:T}$ from a given test set, and we extract their static and dynamic codes, $s^1, s^2$ and $d^1_{1:T}, d^2_{1:T}$. Then, we swap the representations by joining the static code of the first sequence with the dynamic factors of the second sequence, and vice versa. Formally, we generate new samples $\overline{x}^1_{1:T}$ and $\overline{x}^2_{1:T}$ via

$$\overline{x}^1_{1:T} := \text{dec}(s^1, d^2_{1:T}), \quad \overline{x}^2_{1:T} := \text{dec}(s^2, d^1_{1:T}),$$

where dec is the decoder. In Fig. 3, we show two swap examples over the MUG dataset. Additional examples for Sprites and MUG datasets are shown in Figs. 11, 12.

Figure 3: displays a pairwise facial expression swapping experiment. The top left row presents the original facial expressions of two characters. In the row beneath, their expressions have been swapped, demonstrating the outcome of the experiment. This process is replicated with another pair on the right, with the initial and altered expressions shown in the top and bottom rows, respectively. The model adeptly distinguishes static (character) and dynamic (expression) components within each sequence, enabling a seamless swap between the examples.

## 5.3 QUANTITATIVE EVALUATION

### 5.3.1 VIDEO SEQUENCES

**Disentangled generation.** We follow sequential disentanglement works and their protocol to quantitatively evaluate our method and its generative disentanglement features, see e.g., (Zhu et al., 2020; Bai et al., 2021; Naiman et al., 2023). The evaluation process is similar to the swap experiment we detail in Sec. 5.2, where the main difference is that instead of swapping between factors of two sequences, we fix the dynamic features and *sample* the static features from the prior. Specifically, we sample a sequence $x_{1:T}$ from the test set, and we disentangle it to static $s$ and dynamic $d_{1:T}$ factors. Then, we sample a new static code from the prior, which we denote by $\bar{s}$. Finally, we reconstruct the video sequence corresponding to $\bar{x}_{1:T} := \text{dec}(\bar{s}, d_{1:T})$. Naturally, we expect that the reconstruction will share the same dynamic features as the inputs, but will have a different static behavior, e.g., a different person with the same expression.

To quantitatively estimate the new sequences, we use a pre-trained classifier. Essentially, for every reconstructed $\bar{x}_{1:T}$, the classifier outputs its dynamic class, e.g., facial expression in MUG and action in Sprites. We measure the performance of our model with respect to the classification accuracy (Acc), the inception score (IS), intra-entropy $H(u|x)$ and inter-entropy $H(y)$. See App. A.2 for more details on the metrics. We detail in Tab. 1 the results our method obtains, and we compare it to recent SOTA approaches, evaluated on the Sprites and MUG datasets. Similarly to recent works, we achieve $100\%$ accuracy on Sprites, and strong measures on the other metrics (equivalent to SPYL). Further, we report new state-of-the-art results on the MUG dataset with an accuracy of $\mathbf{87.53\%}$, IS $= \mathbf{5.598}$ and $H(y|x) = \mathbf{0.049}$. Consistent with prior works, we do not report the results for fixing the static features and sampling dynamics since all methods get near-perfect accuracy.

**Failure case analysis on MUG.** The MUG dataset is a common disentanglement benchmark, comprising of six expressions made by 52 different subjects. Tab. 1 shows that all models fail to produce $> 90\%$ accuracy for this dataset. In what follows, we explore the failure cases of our model, toward characterizing where and why the model fails. To this end, we observe that *different* facial expressions may look *similar* for different subjects or even for the same subject. For example, we

Table 1: Benchmark disentanglement metrics on Sprites and MUG. Results with standard deviation appear in Tab. 9. Arrows denote whether higher or lower results are better.

| Method | Sprites | | | | MUG | | | |
|---|---|---|---|---|---|---|---|---|
| | Acc↑ | IS↑ | $H(y|x)$↓ | $H(y)$↑ | Acc↑ | IS↑ | $H(y|x)$↓ | $H(y)$↑ |
| MoCoGAN | 92.89% | 8.461 | 0.090 | 2.192 | 63.12% | 4.332 | 0.183 | 1.721 |
| DSVAE | 90.73% | 8.384 | 0.072 | 2.192 | 54.29% | 3.608 | 0.374 | 1.657 |
| R-WAE | 98.98% | 8.516 | 0.055 | **2.197** | 71.25% | 5.149 | 0.131 | 1.771 |
| S3VAE | 99.49% | 8.637 | 0.041 | **2.197** | 70.51% | 5.136 | 0.135 | 1.760 |
| SKD | **100%** | **8.999** | **1.6e−7** | **2.197** | 77.45% | 5.569 | 0.052 | 1.769 |
| C-DSVAE | 99.99% | 8.871 | 0.014 | **2.197** | 81.16% | 5.341 | 0.092 | 1.775 |
| SPYL | 100% | 8.942 | 0.006 | **2.197** | 85.71% | 5.548 | 0.066 | **1.779** |
| Ours | **100%** | 8.942 | 0.006 | **2.197** | **87.53%** | **5.598** | **0.049** | 1.775 |

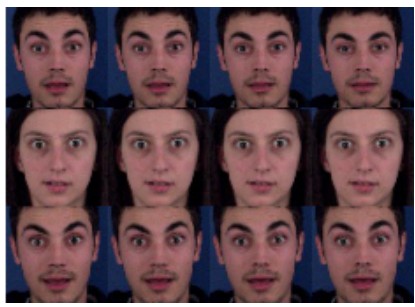 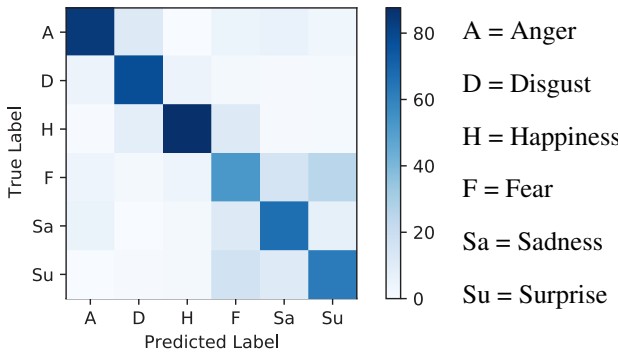

A = Anger

D = Disgust

H = Happiness

F = Fear

Sa = Sadness

Su = Surprise

Figure 4: In analyzing the confusion matrix on the right, it becomes apparent that the model encounters challenges in distinguishing between expressions of fear and surprise, highlighting a notable point of failure. On the characters series on the left, the upper series captures an expression of surprise from $h_1$, while the middle series displays an expression of fear from $h_2$, and the last series shows another instance of fear, this time from $h_1$. These examples underscore the nuanced difficulties the model faces in accurately categorizing emotions, particularly when differentiating between fear and surprise. This subtlety in emotional expression, as demonstrated in the series, can be challenging to discern even with careful human observation.

consider two persons, $h_1$ and $h_2$ and we plot their facial expressions in Fig. 4 (left). The top row is the surprised expression of $h_1$, the middle row is the fear expression of $h_2$, and the bottom row is the fear expression of $h_1$. Clearly, all rows are similar to the human eye, and thus, unsupervised methods such as the one we propose will naturally be challenged by such examples. To quantify this effect we compute the confusion matrix of our model benchmark predictions in Fig. 4 (right). For each cell, we estimate the ratio of predicted labels (columns) vs. the true labels (rows). Thus, the main diagonal represents the true positive rate, whereas off diagonal cells encode false positive and false negative cases. Ideally, we would like to have $100\%$ (dark blue) on the main diagonal, which means that our disentangled generations are perfect. While this is largely the case for Anger, Disgust and Happiness, we find that Fear, Sadness and Surprise may be confounded. For instance, there is $\approx 35\%$ confusion between Fear and Surprise. Our analysis suggests that hierarchical disentanglement based first on dominant features such as subject identity, and only then on facial expression may be a strong inductive bias in modeling disentanglement (Schölkopf et al., 2021). We leave further consideration and exploration of this aspect to future work.

### 5.3.2 TIME SERIES ANALYSIS

In what follows, we evaluate the effectiveness of our approach in disentangling static and dynamic features of time series information. We test the usefulness of the representations we extract for downstream tasks such as prediction and classification. This evaluation methodology aligns with previous work (Oord et al., 2018; Franceschi et al., 2019; Fortuin et al., 2020; Tonekaboni et al., 2022). Our results are compared with recent sequential disentanglement approaches and with tech-

Table 2: TS prediction benchmark.

| | PhysioNet | | ETTh1 |
| **Method** | **AUPRC ↑** | **AUROC ↑** | **MAE ↓** |
|---|---|---|---|
| VAE | $0.157 \pm 0.05$ | $0.564 \pm 0.04$ | $13.66 \pm 0.20$ |
| GP-VAE | $0.282 \pm 0.09$ | $0.699 \pm 0.02$ | $14.98 \pm 0.41$ |
| C-DSVAE | $0.158 \pm 0.01$ | $0.565 \pm 0.01$ | $12.53 \pm 0.88$ |
| GLR | $0.365 \pm 0.09$ | $0.752 \pm 0.01$ | $12.27 \pm 0.03$ |
| SPYL | $0.367 \pm 0.02$ | $0.764 \pm 0.04$ | $12.22 \pm 0.03$ |
| **Ours** | $\mathbf{0.447 \pm 0.02}$ | $\mathbf{0.849 \pm 0.01}$ | $\mathbf{11.39 \pm 0.34}$ |
| RF | $0.446 \pm 0.04$ | $0.802 \pm 0.04$ | $12.09 \pm 0.01$ |

Table 3: TS classification benchmark.

| | PhysioNet | Air Quality |
|---|---|---|
| VAE | $34.71 \pm 0.23$ | $27.17 \pm 0.03$ |
| GP-VAE | $42.47 \pm 2.02$ | $36.73 \pm 1.40$ |
| C-DSVAE | $32.54 \pm 0.00$ | $47.07 \pm 1.20$ |
| GLR | $38.93 \pm 2.48$ | $50.32 \pm 3.87$ |
| SPYL | $46.98 \pm 3.04$ | $57.93 \pm 3.53$ |
| **Ours** | $\mathbf{55.42 \pm 0.23}$ | $\mathbf{62.07 \pm 0.87}$ |
| RF | $62.00 \pm 2.10$ | $62.43 \pm 0.54$ |

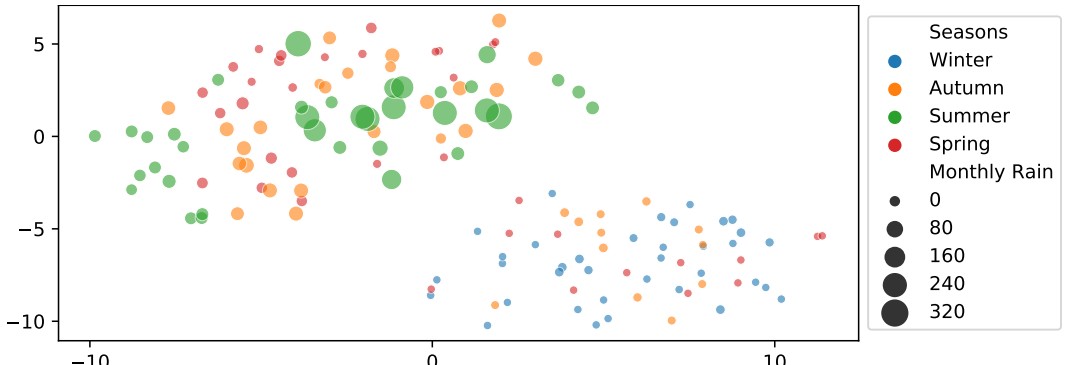

Figure 5: t-SNE visualization of static features from the Air Quality dataset, depicting the model's ability to distinguish days based on precipitation, irrespective of season. Each point represents a day, colored by season and sized by the amount of rain, illustrating the model's nuanced clustering of dry and wet days within the static seasonal context.

niques for time series data such as GP-VAE (Fortuin et al., 2020) and GLR (Tonekaboni et al., 2022). For a fair comparison, we use the same encoder and decoder modules for all baseline methods.

**Downstream prediction tasks.** We consider two scenarios: (i) predicting the risk of in-hospital mortality using the PhysioNet dataset; and (ii) predicting the oil temperature of electricity transformer with ETTh1. In both cases, we train our model on sequences $x_{1:T}$ to learn disentangled static and dynamics features, denoted by $s$ and $d_{1:T}$, respectively. Then, we use the extracted codes as the train set of a simple predictor network. Please see further details on the network structure and hyper-parameters in App. A.3. We compute the area under the receiver operating characteristic curve (AUROC) and area under the precision values curve (AUPRC) on PhysioNet, and the mean absolute error (MAE) for ETTh1, and we report the results in Tab. 2. We also include the baseline results for training directly on the raw features, appears as 'RF' in the table. Remarkably, our method achieves SOTA results and it outperforms all other baseline approaches on the mortality prediction task, including the baseline that trains on the original data. Thus, our results highlight that unlike previous works, our disentangled representations effectively improve downstream prediction tasks–a highly-sought characteristic in representation learning (Bengio et al., 2013; Tonekaboni et al., 2022).

**Static patterns identification.** Reducing error in prediction tasks could be potentially improved by taking into account global patterns (Trivedi et al., 2015), however, the given data samples may include non-useful features. Motivated by this, we explore in this experiment whether our model can properly extract the global patterns via static features we compute in our framework. We follow Tonekaboni et al. (2022) and define a similar evaluation procedure where we choose the ICU unit as the global label for PhysioNet, and the month of the year for the Air Quality dataset. Similar to the above evaluation tasks, we train our model and compute static features. Then, we train a simple multilayer perceptron (MLP) model for classifying the extracted features. We summarize the classification results for both datasets in Tab. 3, and we find that our model better represents static patterns in $s$ in comparison to baseline techniques. In Fig. 5, we project the static code of Air Quality in Beijing using t-SNE. We scale each projected point by the amount of rainfall corresponding to that day, and we color the points by the related season. Our results show a clear clustering of the latent codes with respect to the rainy seasons, that is, the Summer season and its adjacent days from Spring and Autumn are clustered together as opposed to the Winter, which is very dry in Beijing.

### 5.4 ABLATION STUDIES

**Objective function and architecture.** We perform an ablation study to justify our additional loss term and our architectural choice of the subtraction module. This study comprises three specific variations of the method: (1) we train our model without the additional static loss in Eq. 5 (no loss), (2) we train our model without subtracting the static representation from the learned dynamics (no sub), and (3) we train our model without both of them (no both). In all these models, we kept the

Table 4: Ablation study of our model components (top) and its robustness to index choice (bottom).

| Method | AUPRC (PhysioNet) ↑ | AUROC (PhysioNet) ↑ | MAE (ETTh1) ↓ | Accuracy Static (MUG) ↑ |
|---|---|---|---|---|
| no loss | $0.323 \pm 0.022$ | $0.738 \pm 0.017$ | $18.32 \pm 0.324$ | 47.85% |
| no sub | $0.179 \pm 0.007$ | $0.796 \pm 0.013$ | $16.53 \pm 0.134$ | 86.15% |
| no both | $0.325 \pm 0.008$ | $0.654 \pm 0.023$ | $17.42 \pm 0.043$ | 52.15% |
| **Ours** | $\mathbf{0.447 \pm 0.025}$ | $\mathbf{0.849 \pm 0.012}$ | $\mathbf{11.39 \pm 0.3403}$ | **96.12 %** |

| Method | AUPRC (PhysioNet) ↑ | AUROC (PhysioNet) ↑ | MAE (ETTh1) ↓ |
|---|---|---|---|
| $x_{T/2}$ | $0.409 \pm 0.006$ | $0.837 \pm 0.014$ | $11.28 \pm 0.589$ |
| $x_T$ | $0.440 \pm 0.008$ | $0.847 \pm 0.007$ | $11.18 \pm 0.520$ |
| $\mathbf{x_1}$ | $\mathbf{0.447 \pm 0.025}$ | $\mathbf{0.849 \pm 0.012}$ | $\mathbf{11.39 \pm 0.3403}$ |

hyperparameters constant. We show in Tab. 4 the ablation results for PhysioNet, ETTh1 and MUG datasets. Evidently, the removal of components negatively impacts the disentanglement capabilities of our method across multiple tasks. For instance, we observe a significant mode collapse on the MUG dataset, illustrated in low accuracy results for the ablation models, e.g., performance drops by $\approx 10\%$ and $\approx 40\%$ for the no sub and no loss baselines, respectively. Similarly to the MUG case, we also identify a significant performance decrease on the time series tasks.

**Dependence on first sample.** One potential shortcoming of our approach is its dependence on extracting static information from the first sample, see Eq. 4. Here, we would like to empirically verify the robustness of our method to the choice of sample. Thus, instead of using the first sample, we also consider taking the middle $x_{T/2}$ and last $x_T$ samples, i.e., we train models learning the distributions $q(s \mid x_{T/2} ; \phi_s)$ and $q(s \mid x_T ; \phi_s)$. We present in Tab. 4 (bottom) the ablation results on several datasets and tasks. Our results indicate that selecting samples other than the first does not impact the results significantly, highlighting the robustness of our method to the choice of sample.

## 6 LIMITATIONS AND CONCLUSION

Our method is based on a simple heuristic for choosing the first sample as the static source, which may pose a practical limitation and an inherent bias. However, our extensive evaluation and ablation results show that this choice has little effect on the model behavior and overall performance. In particular, we train various models conditioned on samples different from $x_1$ for generating static information, and we find their performance to be similar to models trained on the first sample. Another limitation of our approach, shared by all variational autoencoder techniques, is the quality of generated samples. Essentially, the VAE model inherently balances between reconstruction and regularization, which may negatively impact reconstruction.

In conclusion, sequential disentanglement decomposes input sequences to a single time-invariant component and a series of time-varying factors of variation. Unfortunately, existing variational models suffer from mode collapse issues, where the dynamic factors encode all of the information in the data, and the static code remains non-meaningful. Resolving mode collapse issues via changing the latent subspace dimension and by incorporating mutual information loss terms have been showing limited success. In this work, we observe that the static code may be extracted from a single sample, yielding a new posterior variational model. Further, we design a new neural network that alleviates mode collapse issues by subtracting the extracted static code from the learned dynamic factors. The resulting model is easy-to-code and it has less hyper-parameters in comparison to state-of-the-art approaches. We extensively evaluate our method on standard sequential disentanglement benchmarks including general time series, video and audio datasets. Our model outperforms existing work on generation and prediction tasks as measured by various qualitative and quantitative metrics. In the future, we would like to investigate other architectural backbones such as diffusion models, and we would like to explore sequential disentanglement on real-world datasets and problems.

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

## A  SETUP

### A.1  DATASETS

**MUG.**  Introduced by (Aifanti et al., 2010), MUG encompasses a collection of image sequences featuring 52 subjects exhibiting six distinct facial expressions, namely, anger, fear, disgust, happiness, sadness, and surprise. Each video within the dataset is composed of a variable number of frames, ranging from 50 to 160. To standardize sequence length, as previously demonstrated by (Bai et al., 2021), we employed a procedure in which 15 frames were randomly sampled from the original sequences. Subsequently, Haar Cascades face detection was implemented to isolate facial regions, resizing them to dimensions of $64 \times 64$ pixels. This process resulted in sequences denoted as $x \in \mathbb{R}^{15 \times 3 \times 64 \times 64}$. The post process dataset size is approximately 3500 samples.

**Sprites.**  Introduced by (Reed et al., 2015). This dataset features animated cartoon characters with both static and dynamic attributes. The static attributes present variations in skin, tops, pants, and hair color, each offering six possible options. The dynamic attributes involve three distinct types of motion (walking, casting spells, and slashing) that can be executed in three different orientations (left, right, and forward). In total, the dataset comprises 1296 unique characters capable of performing nine distinct motions. Each sequence within the dataset comprises eight RGB images, each with dimensions of 64 × 64 pixels. We follow previous work protocol and we partitioned the dataset into 9000 samples for training and 2664 samples for testing.

**PhysioNet.**  The PhysioNet ICU Dataset (Goldberger et al., 2000) is a medical time series dataset, encompassing the hospitalization records of 12,000 adult patients in the Intensive Care Unit (ICU). This comprehensive dataset incorporates time-dependent measurements, comprising physiological signals and laboratory data, alongside pertinent patient demographics, including age and the rationale behind their ICU admission. Additionally, the dataset is augmented with labels signifying in-hospital mortality events. Our pre-processing methodology aligns with the protocols outlined in (Tonekaboni et al., 2022).

**Air Quality.**  The UCI Beijing Multi-site Air Quality dataset, as detailed by (Zhang et al., 2017), is a collection of hourly measurements of various air pollutants. These measurements were acquired over a four-year period, spanning from March 1st, 2013, to February 28th, 2017, from 12 nationally regulated monitoring sites. To complement this data, meteorological information for each site has been paired with the nearest weather station under the auspices of the China Meteorological Administration. In alignment with the methodology outlined by(Tonekaboni et al., 2022), our experimental approach involves data pre-processing, entailing the segmentation of samples based on different monitoring stations and months of the year.

**ETTh1.**  The ETTh1 is a subset of the Electricity Transformer Temperature (ETT) dataset, focusing on 1-hour-level data. It contains two years' worth of data from two Chinese counties. The goal is Long Sequence Time-Series Forecasting (LSTF) of oil temperature in transformers. Each data point includes the target value (oil temperature) and 6 power load features. The data is split into train, validation, and test sets, with a 12/4/4-month split ratio.

**Timit**  Introduced by (Garofolo, 1993) the dataset is a collection of read speech, primarily intended for acoustic-phonetic research and various speech-related tasks. This dataset encompasses a total of 6300 utterances, corresponding to approximately 5.4 hours of audio recordings. Each speaker contributes 10 sentences, and the dataset encompasses a diverse pool of 630 speakers, including both adult men and women. To facilitate data pre-processing, we adopt a methodology akin to previous research conducted by (Li & Mandt, 2018). Specifically, we employ spectrogram feature extraction with a 10ms frame shift applied to the audio. Subsequently, segments of 200ms duration, equivalent to 20 frames, are sampled from the audio and treated as independent samples.

A.2 METRICS

**Accuracy** This metric is the common evaluation protocol for assessing a model's capacity to preserve fixed features while generating others. Specifically, it entails the isolation of dynamic features while sampling static ones. This metric is evaluated by employing a pre-trained classifier, referred to as 'C' or the 'judge,' which has been trained on the same training dataset as the model. Subsequently, the classifier's performance is tested on the identical test dataset as the model. For example, in the case of the MUG dataset, the classifier examines the generated facial expression and verifies that it remains consistent during the sampling of static features. We refer this metric along the paper as "Acc" or "Accuracy Dynamic". In addition, we present in the ablation study the metric "Accuracy Static". This metric is exactly the same, just with one small modification. Fixing the static and sampling dynamic.

**Inception Score** ($IS$). This is a metric for the generator performance. First, we apply the judge on all the generated sequences $x_{1:T}$. Thus, getting $p(y|x_{1:T})$ which is the conditional predicted label distribution. Second, we take $p(y)$ which is the marginal predicted label distribution and we calculate the KL-divergence $\text{KL}[p(y|x_{1:T}) \,\|\, p(y)]$. Finally, we compute $IS = \exp\left(\mathbb{E}_x \text{KL}[p(y|x_{1:T}) \,\|\, p(y)]\right)$.

**Inter-Entropy** ($H(y|x)$). Inter-Entropy, often referred to as $H(y|x)$, serves as a metric that reflects the confidence of a classifier ($C$) in making label predictions. A low value of Inter-Entropy indicates high confidence in the predictions made by the classifier. To measure this, we input $k$ generated sequences into the classifier and compute the average entropy over these sequences, given by $\frac{1}{k} \sum_{i=1}^{k} H(p(y|x_{i_{1:T}}))$.

**Intra-Entroy** ($H(y)$). Intra-Entropy, denoted as $H(y)$, is a metric that measures the diversity among generated sequences. A high Intra-Entropy score indicates a high level of diversity among the generated data. This metric is computed by first taking a generated sample from the learned prior distribution $p(y)$ and then applying a judge to obtain the predicted labels $y$. The entropy of this label distribution $H(y)$ quantifies the variability or uncertainty in the generated sequences.

**AUPRC.** The AUPRC (Area Under the Precision-Recall Curve) metric quantifies the precision-recall trade-off by measuring the area under the curve of the precision vs recall. A higher AUPRC indicates better model performance, with values closer to 1 being desirable, indicating high precision and high recall.

**AUROC.** The AUROC (Area Under the Receiver Operating Characteristic Curve) metric quantifies the true positive (TPR) vs false positive (FRR) trade-off by measuring the area under the curve of the those rates. A higher AUPRC indicates better model performance, with values closer to 1 being desirable, indicating high precision and high recall.

**MAE.** The Mean Absolute Error (MAE) metric is a fundamental and widely-used measure in the field of regression analysis and predictive modeling. It quantifies the average magnitude of errors between predicted and actual values, providing a straightforward and intuitive assessment of model accuracy. Computed as the average absolute difference between predicted and true values, MAE is robust to outliers and provides a clear understanding of the model's precision in making predictions.

**EER.** The Equal Error Rate (EER) metric is a vital evaluation measure employed in the context of the speaker verification task, particularly when working with the Timit dataset. EER quantifies the point at which the false positive rate and false negative rate of a model in the speaker verification task are equal. It provides a valuable assessment of the model's performance, specifically in the context of speaker recognition.

## A.3 HYPERPARAMETERS

We compute the following objective function:

$$\mathcal{L} = \max_{\theta, \phi, \psi} \mathbb{E}_{p_D} \left( \mathcal{L}_{\text{recon}} + \mathcal{L}_{\text{reg}} \right) . \tag{9}$$

$$\mathcal{L}_{\text{reg}} = \beta \, \text{KL}[q(s \,|\, x_1 \,;\, \phi_s) \,\|\, p(s)] + \beta \, \text{KL}[q(d_{2:T} \,|\, x_{2:T} \,;\, \phi_d) \,\|\, p(d_{2:T} \,;\, \psi)] , \tag{10}$$

$$\mathcal{L}_{\text{recon}} = \mathbb{E}_{s \sim q_{\phi_s}} \left[ \mathbb{E}_{d_{2:T} \sim q_{\phi_d}} \log p(x_{2:T} \,|\, s, d_{2:T} \,;\, \theta) + \alpha \log p(x_1 \,|\, s, 0 \,;\, \theta) \right] , \tag{11}$$

We determined optimal hyperparameters, specifically $\alpha$ for the reconstruction loss and $\beta$ for the static KL term, using HyperOpt to search for values in the range of Zero to One. It's important to note that we did not normalize the mean squared error (MSE) loss by the batch size during this process. Additionally, optimization was carried out using the Adam optimizer with a learning rate chosen from $0.001, 0.0015, 0.002$, and we considered feature dimensions of $8, 16, 32$ for time series datasets and $64, 128, 256$ for the image and audio datasets for both static and dynamic dimensions. A comprehensive summary of these optimal hyperparameters for each task and dataset is available in Tab.5, and all training processes were limited to a maximum of 1000 epochs.

Table 5: Dataset Hyperparameters

| Dataset | $\alpha$ | $\beta$ | Learning Rate | Batch Size | Static ($s_d$) | Dynamic ($d_d$) |
|---|---|---|---|---|---|---|
| MUG | 0.05 | 0.01 | $1.5 \times 10^{-3}$ | 128 | 128 | 128 |
| PhysioNet | 0.1 | $2 \times 10^{-3}$ | $1 \times 10^{-3}$ | 30 | 8 | 8 |
| Air Quality | 0.2 | $2 \times 10^{-3}$ | $1 \times 10^{-3}$ | 10 | 8 | 8 |
| ETTh1 | 0.1 | $2 \times 10^{-3}$ | $1 \times 10^{-3}$ | 10 | 8 | 8 |
| Timit | 1 | $4 \times 10^{-4}$ | $1 \times 10^{-3}$ | 10 | 256 | 64 |
| Sprites | 0.2 | 0.2 | $2 \times 10^{-3}$ | 128 | 256 | 256 |

## A.4 ARCHITECTURE

In Fig. 1 we present our method architecture. Generally, our architecture comprises of 3 components. The encoder, the disentanglement module and the decoder. The disentanglement module is similar to every data modality and it fully explained in the figure on the main text. The encoder and the decoder for each dataset modality are different, and in what follow, we present the architecture details of each one of them.

**Image:**

1. *Encoder* - The encoder comprises of 5 layers of Conv2d follow by BatchNorm2D followed by LeakyReLU. Below are the Conv2D (input channel dimension, output channel dimension, kernel size, stride, padding) hyperparameters given a $64 \times 64 \times 3$ input image, ordered by the order of their appearance in the encoder: $(3, 32, 4, 2, 1) \rightarrow (32, 64, 4, 2, 1) \rightarrow (64, 128, 4, 2, 1) \rightarrow (128, 256, 4, 2, 1) \rightarrow (256, 128, 4, 2, 1)$.

2. *Decoder* - Similarly to the encoder, the decoder comprises of 5 layer. The first 4 layers are Conv2DTranspose followed by BatchNornm2D followed by LeakyRelu. The final layer is Conv2D followed by BatchNorm2D followed by a Sigmoid function. Below are the Conv2DTranspose (input channel dimension, output channel dimension, kernel size, stride, padding) hyperparameters given a latent code with size of the concatenate static and dynamic $s_d + d_d$ by the order of their appearance in the decoder: $(s_d + d_d, 256, 4, 1, 0) \rightarrow (256, 128, 4, 1, 0) \rightarrow (128, 64, 4, 1, 0) \rightarrow (64, 32, 4, 1, 0) \rightarrow (32, 3, 4, 1, 0)$.

Table 6: Timit voice verification task

| Method | Timit | | |
| | static EER↓ | dynamic EER ↑ | Disentanglement Gap ↑ |
|---|---|---|---|
| FHVAE | 5.06% | 22.77% | 17.71% |
| DSVAE | 5.64% | 19.20% | 13.56% |
| R-WAE | 4.73% | 23.41% | 18.68% |
| S3VAE | 5.02% | 25.51% | 20.49% |
| SKD | 4.46% | 26.78% | 22.32% |
| C-DSVAE | 4.03% | 31.81% | 27.78% |
| SPYL | **3.41%** | 33.22% | 29.81% |
| Ours | 3.83% | **36.75%** | **33.37%** |

**Time Series**

1. *Encoder* - The encoder consists of three linear layers with the next dimensions: $(10, 32) \rightarrow (32, 64) \rightarrow (64, 32)$. ReLU activation's applied after each linear layer.

2. *Decoder* - The decoder is a Linear layer that projects the latent codes into a 32-dimensional space, followed by a tanh activation function. Subsequently, for the different tasks, the output is passed through an LSTM with a hidden size of 32. The LSTM's output is fed into two linear layers, each followed by a ReLU activation: `Linear(32, 64)` and `Linear(64, 32)`. Finally, the output is projected through two more linear layers to produce the mean and covariance parameters, which are used to sample the final output.

# B  ADDITIONAL EXPERIMENTS AND DETAILS

## B.1  AUDIO DATASET - TIMIT

**Experiment description**   To show the robustness of sequential disentanglement methods, previous works has used a common audio verification benchmark on the Timit dataset. We evalute our model using the common benchmark protocol described in (Li & Mandt, 2018) and with the same encoding and decoding architecture for the fairness of comparison. Briefly, the evaluation protocol is given two audio tracks, the goal of this task is to recognize if they come from the same speaker. Unrelated to what the words context are. The verification is done by EER - Equal Error Rate (Chenafa et al., 2008) metric where we use cosine similarity and $\epsilon \in [0, 1]$ threshold. Given two audio tracks $x_{1:T}^1, x_{1:T}^2$ we extract their static $s^1, s^2$ representations. We follow the exact extraction process as in (Bai et al., 2021; Li & Mandt, 2018). Then we measure the cosine similarity of the pairs, if its higher then $\epsilon$ we classify them as the same speaker, else as different speakers. The $\epsilon$ is determined by calibration of the ERR. We conducted the above procedure once for the static features and separately to the dynamic features then we report each experiment EER. In the static setup, lower error is desired, since it should encapsulate information about the speaker. On the other hand, in the dynamic setup the higher the error the better since it should not encapsulate information about the speaker. In addition, we show the disentanglement gap, the gap between the static and dynamic setups, which indicates the quality of the disentanglement. We report our model performance in Tab.6. Notably, our model achieve state-of-the-art Disentanglement Gap, surpassing by approximately $3.5\%$ the best previous method.

## B.2  ABLATION STUDY - SUBTRACTION AND LOSS CONT.

Due to space limitation, we extend here the ablation study results that reported in the main text at Tab.4(top). We report the result on the Physoinet classification task and on the Air Quality classification task In Tab.7. The results support that our model components are crucial for achieving best performance.

Table 7: Ablation study additional results on classification benchmark.

| Method | Accuracy (PhysioNet) ↑ | Accuracy (Air Quality) ↑ |
|---|---|---|
| No-loss | $43.06\% \pm 0.244$ | $52.67\% \pm 0.052$ |
| No-sub | $46.42\% \pm 0.054$ | $55.13\% \pm 0.024$ |
| No-both | $47.42\% \pm 0.026$ | $54.32\% \pm 0.532$ |
| **Ours** | $\mathbf{55.42\% \pm 0.23}$ | $\mathbf{62.07\% \pm 0.87}$ |

Table 8: T frame ablation study additional results on classification benchmark.

| Method | Accuracy (PhysioNet) ↑ | Accuracy (Air Quality) ↑ |
|---|---|---|
| $x_{T/2}$ | $54.80\% \pm 0.140$ | $56.67\% \pm 0.2701$ |
| $x_T$ | $54.29\% \pm 0.760$ | $56.27\% \pm 0.801$ |
| **Ours** | $\mathbf{55.42\% \pm 0.23}$ | $\mathbf{62.07\% \pm 0.87}$ |

Table 9: Sprites and MUG datasets with standard deviation measures.

| | MoCoGAN | DSVAE | R-WAE | S3VAE | SKD | C-DSVAE | SPYL | Ours |
|---|---|---|---|---|---|---|---|---|
| **Sprites** | | | | | | | | |
| Acc↑ | 92.89% | 90.73% | 98.98% | 99.49% | **100%** | 99.99% | $\mathbf{100\% \pm 0}$ | $\mathbf{100\% \pm 0}$ |
| IS↑ | 8.461 | 8.384 | 8.516 | 8.637 | **8.999** | 8.871 | $8.942 \pm 3.3e{-}5$ | $8.942 \pm 7e{-}5$ |
| $H(y\|x)\downarrow$ | 0.090 | 0.072 | 0.055 | 0.041 | **1.6e−7** | 0.014 | $0.006 \pm 4e{-}6$ | $0.006 \pm 3e{-}6$ |
| $H(y)\uparrow$ | 2.192 | 2.192 | **2.197** | **2.197** | **2.197** | **2.197** | $\mathbf{2.197 \pm 0}$ | $\mathbf{2.197 \pm 0}$ |
| **MUG** | | | | | | | | |
| Acc↑ | 63.12% | 54.29% | 71.25% | 70.51% | 77.45% | 81.16% | $85.71\% \pm 0.9$ | $\mathbf{87.53\% \pm 0.9}$ |
| IS↑ | 4.332 | 3.608 | 5.149 | 5.136 | 5.569 | 5.341 | $5.548 \pm 0.039$ | $\mathbf{5.598 \pm 0.068}$ |
| $H(y\|x)\downarrow$ | 0.183 | 0.374 | 0.131 | 0.135 | 0.052 | 0.092 | $0.066 \pm 4e{-}3$ | $\mathbf{0.049 \pm 8e{-}3}$ |
| $H(y)\uparrow$ | 1.721 | 1.657 | 1.771 | 1.760 | 1.769 | 1.775 | $\mathbf{1.779 \pm 6e{-}3}$ | $1.775 \pm 0.013$ |

## B.3 ABLATION STUDY - DEPENDENCE ON FIRST SAMPLE CONT.

Due to space limitation, we extend here the second ablation study results that reported in the main text at Tab.4(bottom). We report the result on the Physoinet classification task and on the Air Quality classification task. In Tab.8 we can see our model robustness for frame choice on more benchmarks and datasets.

## B.4 DISENTANGLEMENT GENERATION WITH STANDARD DEVIATION

For simplicity we present in the main text, in Tab.1 the results without standard deviation. Due to the nature of generative models to produce unstable results its important to validate that a model is stable and show statistically significant improvement. Therefore, we present full result of our model with standard deviation in Tab.9. We repeat the task 300 times with different seeds and report its mean and standard deviation. The results show that our model is profoundly stable.

## B.5 GENERATIVE SAMPLING

In this experiment we show qualitatively our model capability to fix one factor and sample the other on the MUG and Sprites datasets. In Fig.7 and Fig.9 we fix the static component and sample new dynamic component. In Fig.8 and Fig.10 we fix the dynamic component and sample new static component.

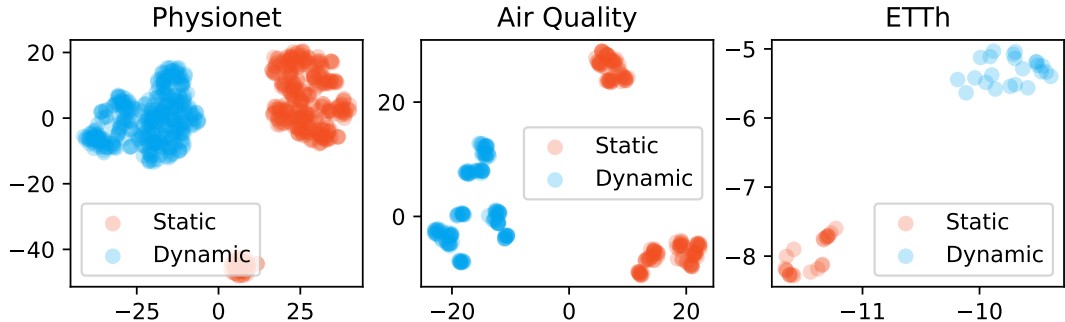

Figure 6: We show clearly separated static and dynamic clusters using t-SNE plots for PhysioNet (left), Air Quality (middle) and ETTh1 (right) of static and dynamic codes.

### B.6 DISENTANGLEMENT FOR STATIC AND DYNAMIC

To assess our model's ability to disentangle data, we conducted an experiment on three time series datasets. We extracted static and dynamic features from each dataset, and visualized them in a two-dimensional space using t-SNE. The clear separation of static and dynamic features in the visualizations, as shown in Fig. 6, indicates our model's ability to effectively separate the static data from the dynamic data.

### B.7 TIME SERIES RECONSTRUCTION

We present a qualitative analysis of our model reconstruction abilities of time series signal. We use the Air Quality datasets for this analysis. This dataset is comprised of multiple features such as Temperature (TEMP), Carbon Monoxide (CO) and other physical environmental features. In Fig.13 each plot represent a different feature. The X axis of each plot is the measurement of a specific measure in a specific day. The Y axis are the measurement values. Notably, we observe from this experiment, that our model successfully captures the semantics of each of the time series features. Which in turn, explicitly imply that the latent features of our model encapsulate valuable information about the observed data.

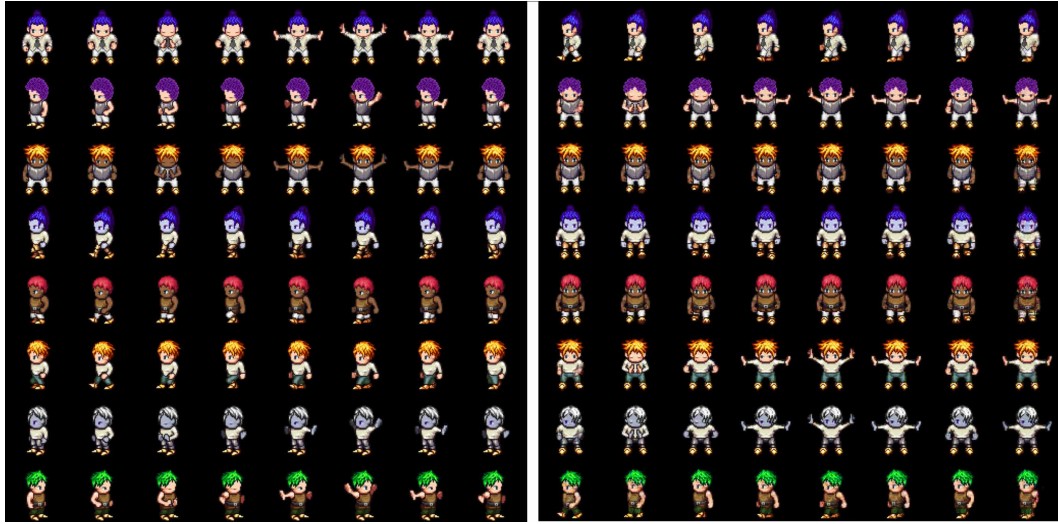

Figure 7: Dynamics generation results in Sprites dataset. On the left side, the original sequence. On the right side, the same sequence with different dynamics.

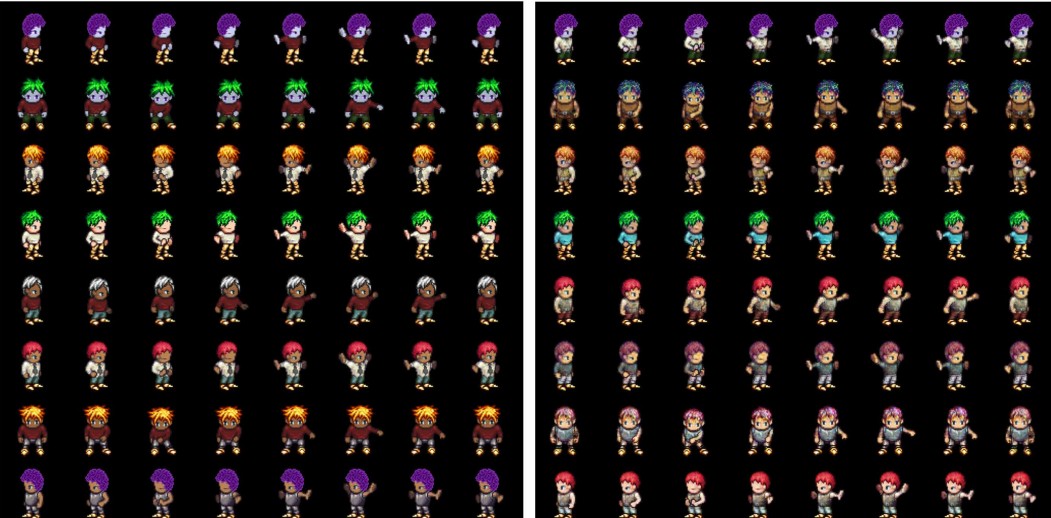

Figure 8: Static generation results in Sprites dataset. On the left side, the original sequence. On the right side, the same sequence with different static features.

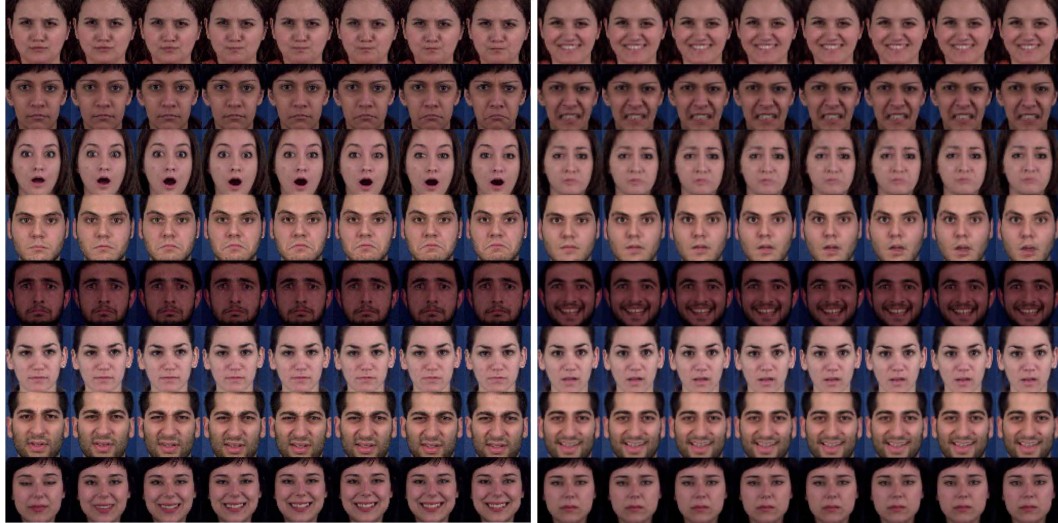

Figure 9: Dynamics generation results in MUG dataset. On the left side, the original sequence. On the right side, the same sequence with different dynamics.

## B.8   SWAP EXAMPLES

We extend the swap experiment on the main text in Sec.5.2 showing more example for swaps between two samples on the MUG and the Sprites in Fig.11 and Fig.12.

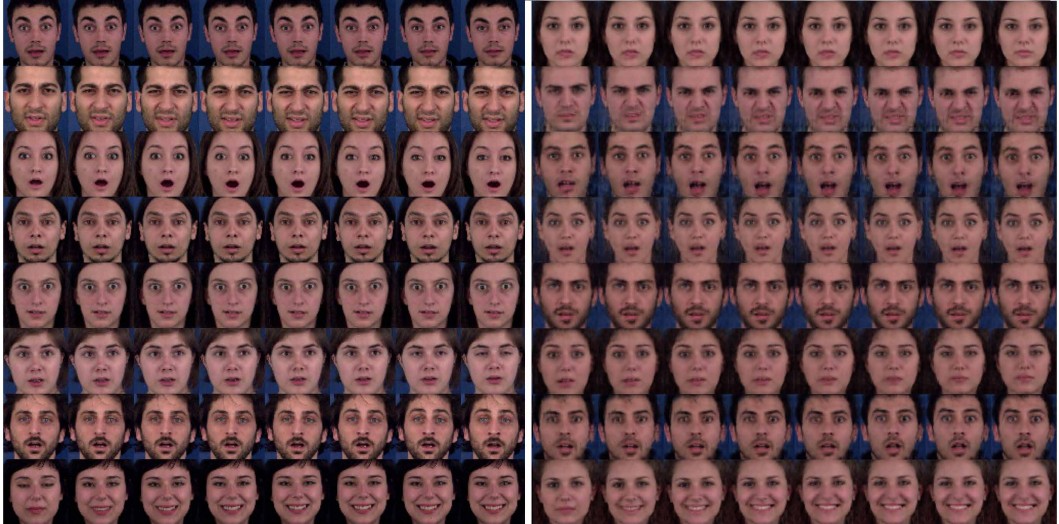

Figure 10: Static generation results in MUG dataset. On the left side, the original sequence. On the right side, the same sequence with different static features.

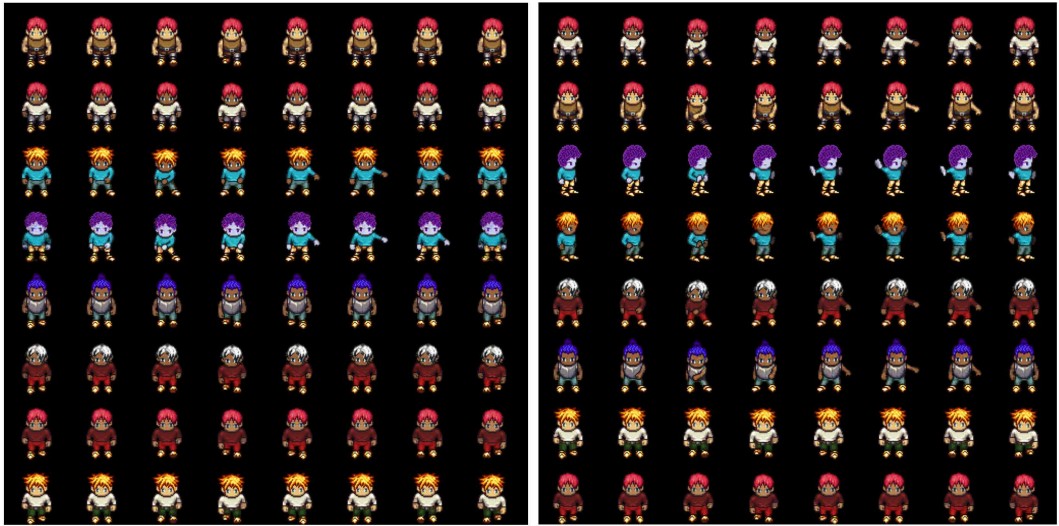

Figure 11: Swapping results in Sprites dataset. Each odd row (counting from One) contains Two original samples. Each even row contains the Two swapped samples of their above row.

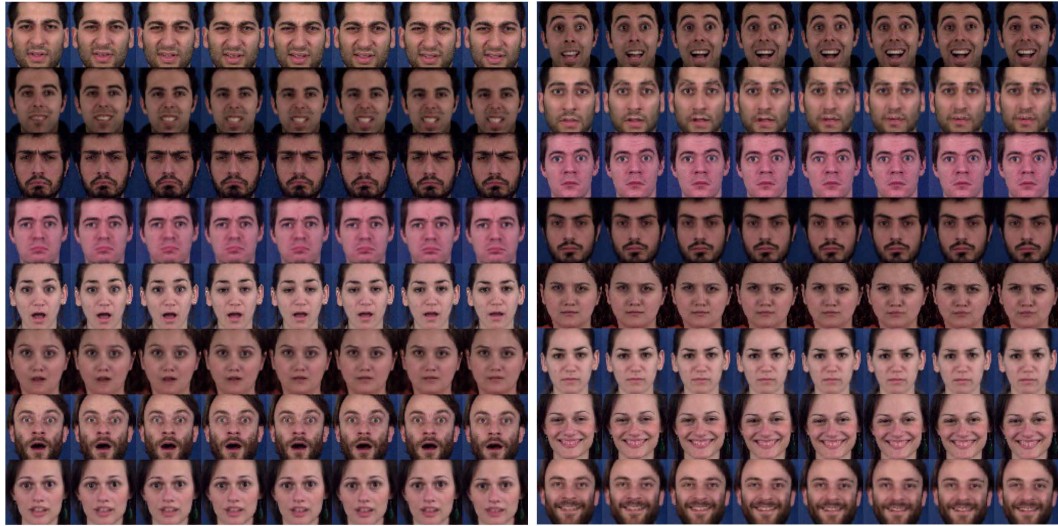

Figure 12: Swapping results in MUG dataset. Each odd row (counting from One) contains Two original samples. Each even row contains the Two swapped samples of their above row.

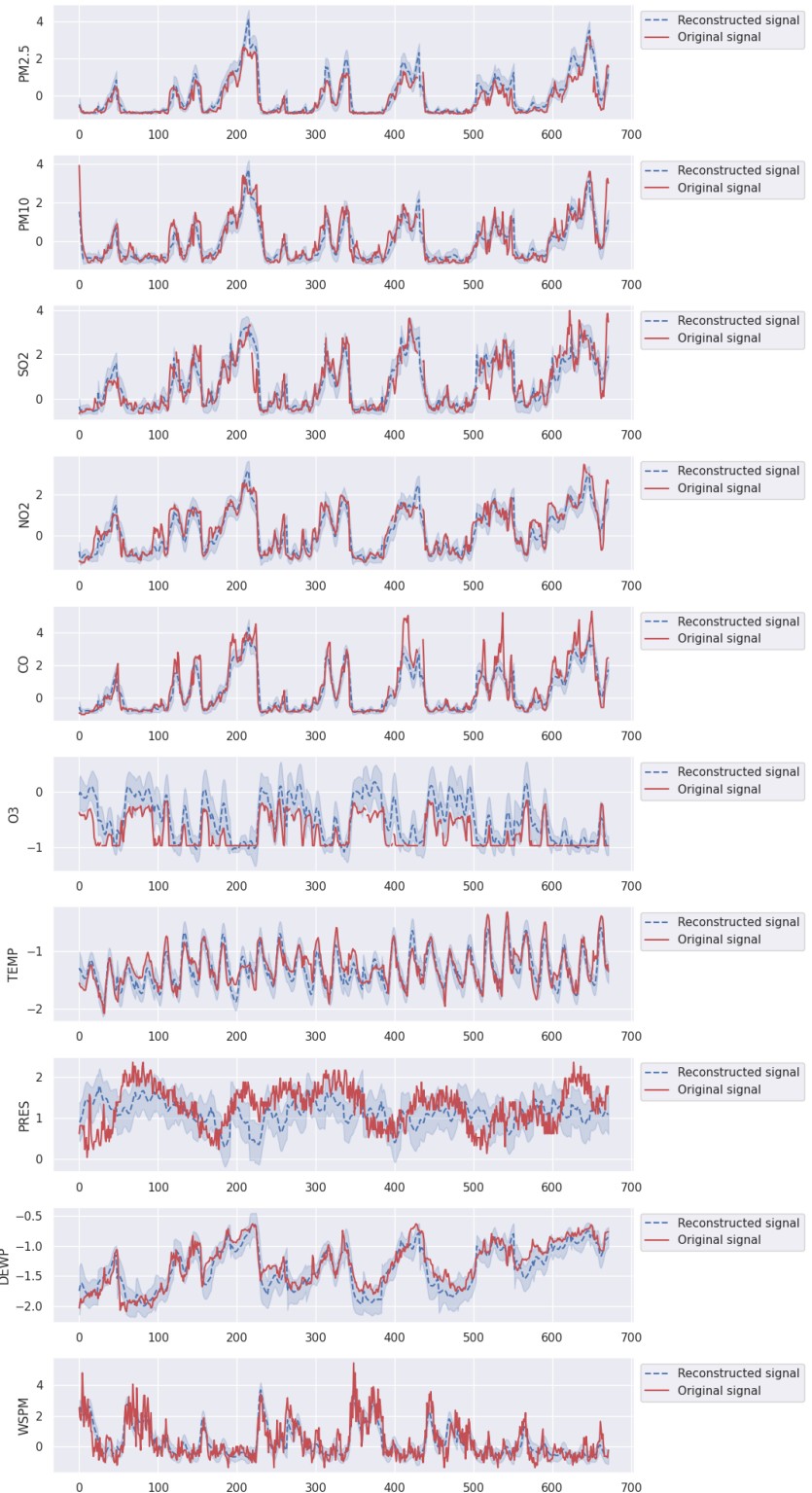

Figure 13: Reconstructed signal of Air Quality by our model

