# OpenReview forum: "Mitigating Mode Collapse in Sequential Disentanglement via an Architecture Bias"
_ICLR.cc/2024/Conference — Submitted to ICLR 2024_

### Official Review · Reviewer_WJJF · 2023-10-28

**Soundness:** 1 poor
**Presentation:** 2 fair
**Contribution:** 1 poor
**Rating:** 1
**Confidence:** 5

**Summary:**

The paper studies the problem of sequential disentanglement in which the goal is to learn representations for two factors that give rise to the observations: i) the time-invariant (static) features that are shared across all images within a sequence; ii) time-varying (dynamic) features that change along the sequence. The authors describe a common failure mode of existing models in which the static features ‘leak’ into the dynamic features representations, and propose a method to encode the static features only from the first observation in a sequence. The approach relies on a model with architecture inductive bias to learn the dynamic features from all the remaining observations. The method is benchmarked on several sequential datasets with different modalities.

**Strengths:**

1. The study of sequential disentanglement is well-motivated as videos are a major modality in real-world data.
2. The paper is well-written and easy to follow.

**Weaknesses:**

1. Technical claims put forth in this paper appear somewhat misleading in my evaluation.
The authors claim that subtracting the static representation from the dynamic representation induces an architectural bias and “removes” static features from leaking into the dynamic features. It is not clear to me why this subtraction may encourage removing static information since the static and dynamic representations are not embedded into the same latent space. In fact, they are not interchangeable when input into the decoder i.e. are not separate full-dimensional latent vectors within a same latent space; rather, these representations are concatenated to form the entire image encoding.
Consequently, I fail to discern any relationship between this architectural approach and a potential inductive bias.

2. Following my previous point, I do not understand why the embeddings of static and dynamic features are being plotted along each other in Fig. 2. According to my understanding, static and dynamic are components / dimensions that form concatenated latent vectors, so what is the meaning of plotting different dimensions within the same embedding space? Why should they align or be distinct? For example, each dimension can follow same gaussian distribution, but still represent different features, as the decoder can apply any function to the latent vector (which is the concatenation of static || dynamic).

**Questions:**

1. The quantitative evaluation includes only image-level metrics (derived from pre-trained classifiers). I believe it would be very valuable to conduct evaluation at the representation-level to measure the disentanglement quality, as the metrics proposed in [1].

2. Could the author compare their method to class-content disentanglement methods as [1]? Class could be treated as the static features and content could be referred to as the dynamic features.

3. I think the concept of ‘Mode collapse’ is often attributed to unconditional generative models and might mislead the reader. It may be better to use a different terminology to describe the degeneration problem in the context of disentanglement.

[1] Gabbay and Hoshen. “Demystifying Inter-Class Disentanglement”. In ICLR, 2020.

---

> ### Author Response · Authors · 2023-11-20
>
> Dear Reviewer WJJF,
>
> Thank you for your thorough review and the critical points you have raised. We value your insights and provide the following clarifications and responses to your concerns:
>
> - We would like to address your observation regarding the latent space of static and dynamic features. In our approach, both feature types are indeed encoded within the same latent space. This shared latent space is fundamental to our model’s design, allowing for the direct comparison and manipulation of static and dynamic features.
> The process we describe, where static features are considered in the computation of dynamic features, is intended to explicitly separate the influence of time-invariant factors from time-variant ones. The operation to “subtract” static from dynamic features is a conceptual simplification. In practice, this is achieved through a designed inductive bias within the model architecture, ensuring that the static features do not “leak” into the dynamic representations. We fully understand that subtracting static content from dynamic factors may seem naive. By subtracting the static representation s, which is less prone to dynamic variation,  we facilitate the LSTM to concentrate solely on dynamic differences, thus streamlining its function. To empirically validate this approach, we included a series of experiments (table 4 and table 8) in the paper. These experiments specifically demonstrate that removing the subtraction operation or the loss term that penalizes the static component’s presence in dynamic factors leads to a significant drop in performance. These results substantiate the necessity and effectiveness of our subtraction method, confirming that it plays a critical role in enhancing the model’s ability to capture and separate time-varying elements accurately. Without this step, the LSTM module’s capacity to discern complete dynamic changes is notably diminished, as reflected in the lower results.
>
> - We appreciate your query regarding the visualization of static and dynamic embeddings in Figure 2. Your point touches on a critical aspect of our methodology—the representation of static and dynamic features within the same dimensional space.
> In our approach, both static and dynamic features are encoded into latent vectors of the same dimensionality. This allows us to compare and contrast these features within the same embedding space, which is crucial for our analysis of disentanglement. The static features, derived from a single frame, capture the invariant aspects of the data, while the dynamic features, obtained from a sequence, encapsulate the time-varying attributes. For visualization purposes in Figure 2, we average the dynamic features across the sequence to reduce them to a single representative vector, comparable in dimension to the static features. We then employ t-SNE, to project these high-dimensional vectors into a two-dimensional space for ease of visualization and interpretation. This method allows us to observe how well the static and dynamic features are separated or entangled. The reason for plotting static and dynamic features together is to demonstrate their relationship and the effectiveness of our model in disentangling them. It is to visually confirm that the static features do not influence the dynamic representation and vice versa, which aligns with our goal of achieving clear disentanglement between invariant and variant factors. We will ensure that the revised manuscript includes a more detailed explanation of the dimensionality and handling of static and dynamic features, as well as the rationale behind their joint visualization.
>
>
> - Representation-Level Evaluation and Comparison with Class-Content Disentanglement Methods:
> We acknowledge the potential insights such a comparison could offer. However, our work is centered on sequential disentanglement, which inherently diverges from static disentanglement in its fundamental assumptions, modeling approach, and the applicable evaluation benchmarks. Sequential disentanglement is tailored to unravel time-related changes and maintain a consistent representation of static features over time, while class-content disentanglement typically addresses static representations where temporal dynamics do not play a role. The methodologies and tools we have developed are thus specifically designed for the challenges and nuances of sequential data. Given this significant conceptual and methodological difference, a direct comparison with class-content disentanglement methods may not adequately reflect the unique contributions and the efficacy of our approach.
>
> - We recognize that the term ‘mode collapse’ is typically associated with unconditional generative models. We used this term to draw an analogy to the concept of representation degeneration; however, we see how this could be confusing. We will consider alternative terminology that more accurately describes the challenges in disentanglement in sequential settings.

---

> > ### Comment · Reviewer_WJJF · 2023-11-23
> >
> > I thank the authors for their response.
> >
> > Although the latent representations of the static and dynamic features are of the same dimension,
> > they are still **concatenated** to construct the latent vector of the image, forming a new latent space. The two separate representations are not embedded in the same latent space and therefore are not interchangeable when input into the decoder.
> >
> > Moreover, it is crucial that any work which make claims towards representation disentanglement will include quantitative analysis at the **representation-level** to show that the two separate encodings represent different features.
> > I do believe that the baselines from class-content disentanglement are totally compatible with the sequential setting presented in this paper and need to be included in the experimental evaluation.
> >
> > For the above reasons, I keep my initial decision.

---

### Official Review · Reviewer_gfeQ · 2023-10-29

**Soundness:** 2 fair
**Presentation:** 3 good
**Contribution:** 2 fair
**Rating:** 5
**Confidence:** 4

**Summary:**

This work presents a sequential disentanglement model designed to separately learn the static component and dynamic factors. The model subtracts the static content from the series of learned dynamic factors. The authors then evaluate its performance on various data-modality benchmarks.

**Strengths:**

- The manuscript is written clearly and is easy to follow.
- The motivation of static posterior is conditioned on a single series element is reasonable. The implementation of learning the static factor independently of the sequence is reasonable.
- The authors conduct extensive experiments to evaluate the effectiveness.

**Weaknesses:**

- I am worried about how to ensure that s contains only static features. The authors claim that static factors can be extracted from a single frame in the sequence, which is not a necessary and sufficient condition. Otherwise, any frame from the video can be used. Why the first frame?
- In addition, in Equation 8, if s contains dynamic factors, subtracting s from the dynamic information may result in the loss of some dynamic information, making it difficult for the LSTM module to capture the complete dynamic changes.
- The method of removing static information from dynamic information is by subtraction between features, which is quite naive.

**Questions:**

- How to ensure that small time-varying vectors are captured?
- Does the length of the time series affect the performance of the model?

---

> ### Author Response · Authors · 2023-11-20
>
> Dear Reviewer gfeQ,
>
> Thank you for your thoughtful review and for highlighting the clarity of the manuscript as well as the rationale behind our approach to disentangle static and dynamic factors.
>
>
> - We leverage the fact that in sequential data, a single frame in a series has no time-dependent information. Therefore, a single frame will not encode dynamic features but only static features. Ablation experiments were conducted to assess the impact of utilizing different frames, and the results unequivocally demonstrated that employing the initial frame yields the most favorable outcome. We acknowledge the scenario you described, where the first data point might be more entangled, could be problematic. To mitigate this, we propose an additional check of different frames for the static factors, which we included in our paper. Our results (table 4 and table 8) show that even in cases where the first frame may not be ideal, our method still successfully disentangles the factors, as evidenced by the strong performance across different checkpoints in a sequence. This suggests that the model does not merely rely on the first frame but rather learns a robust representation of time-invariant features that are consistent across the entire sequence.
>
>
> - Regarding the subtraction method outlined in Equation 8, your concern is valid. if $s$ erroneously contains dynamic information, this could indeed lead to a loss of dynamic content. To mitigate this risk, our model is designed to ensure that $s$ is as pure a representation of static features as possible. We acknowledge your concern regarding the subtraction methodology. To validate its efficacy, we conducted experiments (table 4), detailed in our paper, that compare the performance with and without the subtraction of the static component (ablation study).  These experiments clearly show that excluding the subtraction step significantly reduces the model’s performance, highlighting the method’s importance in achieving effective disentanglement of static and dynamic features.
>
>
> - We understand your concern that subtracting the static part from the dynamic might sound too naive. We believe that the simplicity we bring is what makes our method unique and the empirical experiments (table 4) we performed indeed prove that this subtraction is significant to the results of the model on the benchmarks.
>
> - Small time-varying vectors are captured by the dynamic LSTM module, which is trained to focus on changes between frames after the static component has been subtracted. This is designed to be sensitive to even minimal dynamic variations.
>
> - About your question about the length of the time series affecting the model’s performance, we have indeed tested our model on datasets with varying sequence lengths and found that our approach consistently achieves state-of-the-art performance across all of them. At the same time, since the perception of the dynamic parts in our model depends on LSTM, for particularly long series the model may fail due to the use of the LSTM network.
>
> We hope these explanations address your concerns, and we appreciate the opportunity to clarify these aspects of our work. We believe our method contributes meaningfully to the field of disentangled representation learning, and we are open to further questions or suggestions you might have.

---

> > ### Comment · Reviewer_gfeQ · 2023-11-23
> > **Response**
> >
> > Thanks for your response. Some of my comments have been addressed, so I'll bump up the score.

---

### Official Review · Reviewer_MtBA · 2023-10-31

**Soundness:** 2 fair
**Presentation:** 3 good
**Contribution:** 2 fair
**Rating:** 6
**Confidence:** 4

**Summary:**

In this paper, the authors try to learn disentangled representations for time series data with time-invariant factors and time-dependent factors, by proposing to define a specific form of approximate variational posterior. Specifically, they encode the first data point in a time series into the time-invariant factors, such that they can "subtract" it from the subsequent data points. The authors claim that this can improve the disentanglement between these two types of factors of variation.
Empirically they evaluate their method on multiple time series datasets and compare with different baselines, and they show that their approach outperform the baselines.

**Strengths:**

Their work is well motivated in the sense that it is challenging to disentangle time-invariant factors and time-dependent factors on time series data, particularly on high-dimensional data (e.g., videos). The paper is well-structured and clearly written. The math in the methodology section is detailed and sound, to the best knowledge. Finally, they provide sufficient experimental results in the evaluation.

**Weaknesses:**

I don't fully understand why we would want to have the time-invariant factors encoded merely from one single data point in a time series. Intuitively, the time-invariant and time-dependent factors are entangled at each time step, and there are time steps where those factors are more difficult to disentangle. For example, one video frame where 2 moving objects are highly cluttered while we want to disentangle their appearances (time-invariant) from their locations (time-dependent). Thus I would imagine that it is actually more reasonable to get some form of "average" across all time steps because that "average" is the part that does not change across time and, hence is the time-invariant factor. I understand that prior work sometimes has challenges in learning such disentangled representations for high-dimensional data, but I personally don't think merely picking up one single data point and encoding it into time-invariant can resolve this problem in general. Consider that case where the first data point is the more entangled data point across the whole time series, then you wouldn't get good representations at the beginning, thus the conditioning in the posterior of time-dependent factors would introduce bias. This is my main concern of the method in this paper.

**Questions:**

I don't have technical questions so far. My main question was in the previous section.

---

> ### Author Response · Authors · 2023-11-20
>
> Dear Reviewer MtBA,
>
> Thank you for your detailed review and for acknowledging the motivation and structure of our paper. We appreciate your feedback on the challenge of disentangling time-invariant and time-dependent factors in time series data, particularly in high-dimensional datasets like videos.
>
> - We leverage the fact that in sequential data, a single frame in a series has no time-dependent information. therefore, a single frame will not encode dynamic features but only static features. Ablation experiments were conducted to assess the impact of utilizing different frames, and the results unequivocally demonstrated that employing the initial frame yields the most favorable outcome. We acknowledge the scenario you described, where the first data point might be more entangled, could be problematic. To mitigate this, we propose an additional check of different frames for the static factors, which we included in our paper. Our results (table 4 and table 8) show that even in cases where the first frame may not be ideal, our method still successfully disentangles the factors, as evidenced by the strong performance across different checkpoints in a sequence. This suggests that the model does not merely rely on the first frame but rather learns a robust representation of time-invariant features that are consistent across the entire sequence.
>
>
>  - Our empirical experiments reveal that employing the 'average' encoding method leads to the incorporation of undesirable dynamic features, exacerbating the mode collapse problem. The current model architecture, involving arithmetic subtraction operations, inadvertently discards essential 'dynamic' information, resulting in unintended information leakage. In contrast, when coding static information using a single frame within our architecture, we ensure the exclusion of dynamic information. This approach offers a more effective solution, mitigating the risk of information leakage and addressing the limitations associated with the 'average' encoding technique.
>
>
> - Our approach involves independently establishing a representation for the initial frame, irrespective of its level of entanglement with subsequent frames. Regardless of the degree of entanglement, we consistently learn the static aspect, with the dynamic aspect consistently determined to be zero for the chosen frame. Consequently, it can be inferred that the remaining frames are encoded with respect to the selected anchor frame. This method streamlines the learning process by allowing the network to concentrate on coding the dynamic component, as the static part has already been encoded. Moreover, our model exhibits superior results compared to other models, as demonstrated by our findings.
>
> We believe that these revisions will substantially improve the manuscript and provide a clearer understanding of our model’s contributions. We thank you once again for your valuable feedback.

---

### Official Review · Reviewer_nhge · 2023-11-01

**Soundness:** 4 excellent
**Presentation:** 4 excellent
**Contribution:** 2 fair
**Rating:** 6
**Confidence:** 4

**Summary:**

The authors propose an approach to learning disentangled representations while mitigating mode collapse in sequential datasets. They propose an approach by which models extract static and dynamical properties independently by anchoring the former in the initial input in the sequence processing the latter inputs as an offset of the former in representation space. The introduce latent penalties to accommodate for this modification and encourage sequential disentanglement. They then proceed to extensively test their proposal on a number of datasets while comparing it with previous work.

**Strengths:**

1. The model is well motivated based on limitations of previous work.
2. The description of the model and the derivation of novel penalties is clear and easy to follow.
3. Potential issues are described and the authors even address why their method fails to surpass a performance threshold on one specific dataset.
4. To the best of my knowledge they compare against relevant approaches.
5. The model is tested on a substantial amount of datasets.

**Weaknesses:**

1. Some figures are not very clear. For example in Figure 4 what is swapped? Is it the face or the expression? The caption should contain this information.
2. Some of the results for example in Table 1 make the model seem way less impressive. While indeed the model surpasses all the others, the increase in performance is somewhat small. Thus I don't think being SotA is the most compelling argument for this approach.
3. Similarly, there doesn't seem to be much disentanglement of the different properties in the Air Quality dataset.

**Questions:**

1. Given that the authors stress the importance of mitigating mode collapse as an advantage of their model, why is there no comparison on how prevalent this issue is between their model and the alternatives? Given that as I have previously said the improvement in performance is not that great, then this would be an alternative way to support the idea that this is indeed a substantial improvement over previous approaches.
2. Similarly, I would like to see latent plots of alternative models as a way to compare the disentanglement properties of this approach.

---

> ### Author Response · Authors · 2023-11-20
>
> Dear Reviewer nhge,
>
> Thank you for your thoughtful review and the insights you have provided. We are grateful for the positive remarks about the motivation and clarity of our model, as well as the extensive testing conducted. We also appreciate your constructive feedback, which we address below:
>
> - We acknowledge that the clarity of Figure 4 can be improved. We intended to illustrate the swapping of expressions while keeping the identity constant. To clarify, we will revise the figure caption to explicitly state what elements are swapped and ensure that the visual representation is more apparent.
>
> - Thank you for pointing out the performance improvements shown in Table 1. In the paper we detail under 'Failure case analysis on MUG' why this dataset is difficult because some of the facial expressions, such as surprise and fear, look very similar. It is hard to tell them apart, even for people. Thus, little improvements are important. This means our model is really good at spotting the tiny differences in these expressions. We will take care to explain this better in our article, and see why these small steps forward are important.
>
> - Figure 6 demonstrates the model’s ability to autonomously detect and categorize patterns within the static seasonal context of the Air Quality dataset, notably distinguishing between days with different rainfall levels. While seasonality provides a consistent background throughout the year, our model is not limited by this uniformity. It adeptly clusters days with similar weather patterns, such as grouping rainy days, which is significant as it was achieved without explicit instruction to do so. This self-guided clustering, evident in the figure, showcases the model’s nuanced comprehension of static data components. We will improve figure 6 description in our paper.
>
> - We acknowledge its importance and in our current work, we have primarily relied on theoretical foundations and qualitative analysis to demonstrate our model's capability in mitigating mode collapse. While further empirical comparisons could be insightful, we believe the existing evidence sufficiently supports our claims. We will consider such comparisons in future research.
>
> - We would like to point out an advantage of our model that is relevant to this discussion: our model's flexibility in defining the dimensions of static and dynamic components. Contrasting with previous methods that allocate different dimensions for static and dynamic features, our model designates identical dimensionality to both. This consistent dimensionality is what permits us to apply t-SNE for projecting into a two-dimensional space, enabling the clear comparative visualizations showcased in our work. We would like to point out an advantage of our model that is relevant to this discussion: our model's flexibility in defining the dimensions of static and dynamic components. Other methods developed before us give different dimensions to the static component and the dynamic component. Unlike them, we gave the static factor and the dynamic factor the same dimension, which allows us to use tsnt to project the latent space into two dimensions and present it as seen in our work.
>
> We believe that these revisions will substantially improve the manuscript and provide a clearer understanding of our model’s contributions. We thank you once again for your valuable feedback.

---

> > ### Comment · Reviewer_nhge · 2023-11-22
> >
> > I thank the authors for their detailed response.
> >
> > On the one hand, I am satisfied with some of the responses (e.g. that the proposed model allows for projecting a joint representation of the static and dynamic factors). Unfortunately, I still think that the empirical proof that this approach is better at preventing mode collapse is insufficient.
> >
> > I will raise my score slightly as a result.
> >
> > PS: Do fix the captions, the are not very polished.

---

### Meta-Review · Area_Chair_KTBs · 2023-12-10

**Metareview:**

This paper proposes to learn disentangled representations in terms of both time-variant and time-invariant components for sequential data. While the motivation is clear, some reviewers still held crucial concerns after the rebuttal. The main concern is that there is not clear support that the key design of "feature subtraction" based on the first frame can induce an architectural bias and encourage the desired disentanglement. Also, the "disentanglement" claim has not been fully analyzed at the representation level. These impose alerts that not all claims in the paper are fully supported by empirical studies.

**Justification For Why Not Higher Score:**

All initial reviews were negative. Two reviews were bumped up to "6" based on the rebuttal. Other reviewers were not convinced by the rebuttal, especially in the "disentanglement" misclaimed by the paper, as well as the unsatisfactory architectural design.

**Justification For Why Not Lower Score:**

N/A

---

### Decision · Program_Chairs · 2024-01-16

Reject